# Does Sex Dimorphism Exist in Dysfunctional Movement Patterns during the Sensitive Period of Adolescence?

**DOI:** 10.3390/children7120308

**Published:** 2020-12-20

**Authors:** Josip Karuc, Mario Jelčić, Maroje Sorić, Marjeta Mišigoj-Duraković, Goran Marković

**Affiliations:** 1Faculty of Kinesiology, University of Zagreb, Horvaćanski zavoj 15, 10000 Zagreb, Croatia; maroje.soric@kif.unizg.hr (M.S.); marjeta.misigoj-durakovic@kif.unizg.hr (M.M.-D.); goran.markovic@kif.unizg.hr (G.M.); 2Motus Melior, Sport and Rehabilitation Center, Hektorovićeva ulica 2, 10000 Zagreb, Croatia; mjelcic1@gmail.com

**Keywords:** FMS^TM^, pubescence, maturation, fundamental movement patterns, functional movement, gender difference

## Abstract

This study aimed to investigate sex difference in the functional movement in the adolescent period. Seven hundred and thirty adolescents (365 boys) aged 16–17 years participated in the study. The participants performed standardized Functional Movement Screen™ (FMS^TM^) protocol and a t-test was used to examine sex differences in the total functional movement screen score, while the chi-square test was used to determine sex differences in the proportion of dysfunctional movement and movement asymmetries within the individual FMS^TM^ tests. Girls demonstrated higher total FMS^TM^ score compared to boys (12.7 ± 2.3 and 12.2 ± 2.4, respectively; *p* = 0.0054). Sex differences were present in several individual functional movement patterns where boys demonstrated higher prevalence of dysfunctional movement compared to girls in patterns that challenge mobility and flexibility of the body (inline lunge: 32% vs. 22%, *df* = 1, *p* = 0.0009; shoulder mobility: 47% vs. 26%, *df* = 1, *p* < 0.0001; and active straight leg raise: 31% vs. 9%, *df* = 1, *p* < 0.0001), while girls underperformed in tests that have higher demands for upper-body strength and abdominal stabilization (trunk stability push-up: 81% vs. 44%, *df* = 1, *p* < 0.0001; and rotary stability: 54% vs. 44%, *df* = 1, *p* = 0.0075). Findings of this study suggest that sex dimorphisms exist in functional movement patterns in the period of mid-adolescence. The results of this research need to be considered while using FMS^TM^ as a screening tool, as well as the reference standard for exercise intervention among the secondary school-aged population.

## 1. Introduction

Physical inactivity represents a global health problem and is related to higher risk for morbidity and mortality [1]. Evidence has shown that inactive children are exposed to increased cardiometabolic risk [2,3]. Physical activity in childhood and adolescence is important to attain appropriate bone mineral content [4]. Although the influence of physical activity as a measure of movement quantity has been examined extensively, very few studies have examined the movement quality through the sensitive period of adolescence. However, these studies pointed out the importance of proper development of the optimal functional movement patterns through adolescence [5,6,7,8,9,10,11,12,13,14,15,16]. Since functional movement is considered the clinical measure of movement quality [17,18] and potentially the essential component for optimal motor development, the investigation of the optimal functional movement in childhood and adolescence needs special attention. 

Functional movement can be defined as optimal flexibility of the soft tissue, mobility of the joints, and neuromuscular control of the body regions involved in the particular motor task [17,18]. On the other hand, dysfunctional movement (DFM) is characterized by movement compensations evident across the kinematic chain with a significant loss in mobility, observed asymmetry, and poor movement control of the particular motor task [17,18]. The importance of functional movement patterns has been studied widely [19,20,21] and they represent the basic foundation for the execution of more complex motor tasks [17,18]. A higher incidence of musculoskeletal injury has been associated with DFM patterns among the athletic population [19,20,21], while some studies reported the opposite [22,23,24]. The most common diagnostic tool to assess functional movement is Functional Movement Screen (FMS^TM^) which evaluates mobility and stability in seven functional movement patterns: deep squat, hurdle step, inline lunge, shoulder mobility, active straight leg raise (ASLR), trunk stability push-up, and rotary stability [17,18]. FMS^TM^ can detect movement asymmetries if a difference between the right and left side of the uni/contralateral movement patterns is observed [17,18]. What is more, the literature shows that movement asymmetries detected via FMS^TM^ have been associated with higher injury risk [5] which could possibly contribute to the development of musculoskeletal deformities in later life. 

The presence of the DFM patterns and movement asymmetries in childhood could facilitate postural abnormalities in the period of mid-adolescence. Indeed, evidence shows that neuromuscular control of the movement is not properly developed by the time of the adolescent period [25]. Therefore, identifying DFM patterns and movement asymmetries in this period of a child’s growth needs special attention. Still, only a few studies have investigated sex differences in functional movement in an average or athletic adolescent population. These studies suggest that, in both the general and athletic population, girls exhibit better functional movement compared to boys [8,9,10,12,15,16], while some studies reported opposite or no difference between sexes [6,11,13,14]. However, these were either small-scale studies [11,12,13] or included only active adolescents [5,6,7,8,9,10,14] or adolescents with overweight/obesity [7] and did not analyze movement asymmetries. 

However, to this date, none of the studies have investigated sex differences in functional movement and movement asymmetries in a large representative sample of school-aged mid-adolescents. Therefore, the purpose of this study was to examine sex dimorphism in functional movement patterns and movement asymmetries in the representative sample of mid-adolescents. 

## 2. Materials and Methods

### 2.1. Participants

This investigation is a part of the Croatian physical activity in adolescence longitudinal study (CRO-PALS) conducted in a representative sample of urban youth (city of Zagreb, Croatia). This study was performed during the second wave of assessments, and all measurements were taken in 2015, during March, April, and May. Information about the procedures of the CRO-PALS longitudinal study have been documented in previous research [26]. In brief, using stratified two-stage random sampling procedures (school level and class level), 54 classes in 14 secondary schools were selected to participate in the CRO-PALS study (schools were stratified by type: grammar schools/vocational schools/private schools). All 1408 students in the selected classes were approached, and 903 agreed to participate (response rate = 64%). One hundred and twenty participants were unavailable on the day of testing or did not complete the FMS^TM^ screening. Of one hundred and twenty participants, one hundred and seventeen were unavailable on the day of testing because they were missing from the school at the time of the measurements, whereas three subjects did not complete FMS^TM^ screening due to lack of time (1 girl and 2 boys). As a consequence, we included data from 783 adolescents. All the participants had to meet certain criteria for the medical doctor to perform the screening process, specifically: (1) not having any pain during the movement screening (i.e., FMS^TM^ testing procedures), (2) not having an acute medical condition that precluded FMS^TM^ testing (neurologic disorders or serious orthopedic trauma such as bone fractures or complete muscle ruptures). Accordingly, 53 subjects were excluded. Therefore, the total number of participants that were analyzed was 730 (girls, *n* = 368, mean age ± SD = 16.6 ± 0.4 years old (yo), mean weight ± SD = 60.1 ± 9.3, mean height ± SD = 166.3 ± 6.4; boys, *n* = 362, mean age ± SD = 16.7 ± 0.4 yo, mean weight ± SD = 71.7 ± 12.5, mean height ± SD = 179.0 ± 7.2). The flowchart of the included participants is shown in Figure 1.

Children and their parents were fully informed about the purposes of the research, its protocols, and possible hazards and discomforts related to the procedures used. Written consent was obtained from both children and their parents or legal guardians. The study was performed according to the Declaration of Helsinki and the procedures were approved by the Ethics Committee of the Faculty of Kinesiology, University of Zagreb (No: 1009-2014). 

### 2.2. Functional Movement Screen 

FMS^TM^ is an instrument designed for the evaluation of mobility and stability of seven functional movement tests: the deep squat, hurdle step, inline lunge, shoulder mobility, ASLR, trunk stability push-up, and rotary stability [17,18]. In the current study, ten novice trained raters used FMS^TM^ according to the official guidelines. All ten raters passed a two-day FMS^TM^ education course by an FMS^TM^ certified practitioner. Despite a large number of raters recruited in this study, previous research reported moderate to good interrater and intra-rater reliability of the FMS^TM^ among novice raters [27,28]. Participants had a maximum of three trials for each test in accordance with the recommended protocol [17,18] while each test was scored on a four-point scale, from 0 to 3, with higher scores indicating better functional movement. Evidence shows that pain can alter movement control [29]. Therefore, subjects were asked if they felt pain during the FMS^TM^ assessment and were subsequently scored with a score of 0 and excluded if they answered this question positively (*n* = 53). In the current study, a functional movement was defined as the movement with a given score of 2 or 3 during FMS^TM^ testing. Also, a score of 1 was recorded when the participant was unable to perform the movement task due to the number of movement compensations present, which reflects the DFM pattern [17,18]. This means that a score of 2 and 3 was an indicator of functional movement, whereas a score of 1 was an indicator of DFM for each of the 7 individual FMS^TM^ tests. If a discrepancy in the scores between the right and left side of the contra/unilateral FMS^TM^ test was observed, movement asymmetry was documented for that specific FMS^TM^ test. We analyzed movement asymmetries for five contra/unilateral FMS^TM^ tests (i.e., hurdle step, inline lunge, shoulder mobility, ASLR, and rotary stability). Accordingly, number (n) and proportion (%) of subjects who performed DFM or showed movement asymmetry could be calculated in each of the seven or five individual FMS^TM^ movement patterns, respectively. This was the basic step for analyzing the differences in the proportion of participants who performed DFM or demonstrated any asymmetry between girls and boys for individual FMS^TM^ tests (i.e., using chi-square tests). In addition, the total FMS^TM^ score was set as an outcome continuous variable and was calculated according to the literature [17,18].

### 2.3. Sport Participation

In order to assess whether someone participated in an organized sport activity or not, the questionnaire was offered with two YES/NO questions inquiring about regular participation in organized sports in school, as well as outside of the school. For participants who stated that they participated in organized sport, a comprehensive list of sports activities was offered and participants identified all the sports in which they regularly participated. 

### 2.4. Statistical Analysis 

An independent *t*-test was used to examine differences between sexes in total FMS^TM^ score. Chi-square test was performed to investigate differences between girls and boys in the proportion of DFM in 7 individual FMS^TM^ tests and for the movement asymmetries exhibited in the 5 contralateral FMS^TM^ tests. In addition, the same analyses concerning sex differences were done for the group of non-athletic and athletic participants separately. Data are presented as mean ± SD. All analyses were performed using Statistica (version 13.0) and the level of statistical significance was set at *p* < 0.05.

## 3. Results

The basic characteristics of participants are shown in Table 1. Results demonstrated that girls slightly outperformed boys in total FMS^TM^ score (12.7 ± 2.4 and 12.2 ± 25, respectively; *p* = 0.0054).

Figure 2 depicts the proportion (%) of DFM patterns among girls and boys in all seven FMS^TM^ tests. Girls demonstrated a higher proportion of DFM patterns compared to boys in trunk stability push-up (81% vs. 44%, *df* = 1, *p* < 0.0001) and rotary stability (54% vs. 44%, *df* = 1, *p* = 0.0075). However, boys showed a higher proportion of DFM in inline lunge (32% vs. 22%, *df* = 1, *p* = 0.0009), shoulder mobility (47% vs. 26%, *df* = 1, *p* < 0.0001), and ASLR (31% vs. 9%, *df* = 1, *p* < 0.0001), while scores in deep squat and hurdle step were similar in both sexes (see Figure 2).

Boys demonstrated a higher proportion of movement asymmetries compared to girls in shoulder mobility (45% vs. 36%, *df* = 1, *p* = 0.0218) and ASLR (21% vs. 13%, *df* = 1, *p* = 0.008). However, no significant difference between girls and boys in the proportion of the movement asymmetries was found for the other FMS^TM^ tests: hurdle step (27% vs. 24%, *df* = 1, *p* = 0.331), inline lunge (31% vs. 31%, *df* = 1, *p* = 0.95), and rotary stability (26% vs. 22%, *df* = 1, *p* = 0.237) (see Figure 3).

### Subgroup Analyses

Since five subjects did not report sport participation status (due to being absent from school), all subgroup analyses were based on 725 participants. When the sample was stratified by sport participation, within the group of non-athletic participants there was no sex differences in the total FMS^TM^ score (girls vs. boys, 12.6 vs. 12.2, respectively, *p* = 0.11). However, among the athletic subgroup of adolescents, girls significantly outperformed boys (13.2 vs. 12.3, *p* = 0.002). Among the non-athletic adolescents, boys demonstrated higher proportion of DFM in three tests compared to girls (inline lunge: 24% vs. 33%, *df* = 1, *p* = 0.032; shoulder mobility: 29% vs. 41%, *df* = 1, *p* = 0.004; ASLR: 7% vs. 34%, *df* = 1, *p* < 0.0001). On the other hand, girls underperformed in push-up and rotary stability tests (push up: 80% vs. 44%, *df* = 1; *p* < 0.0001; rotary stability: 56% vs. 47%, *df* = 1; *p* = 0.04), while in squat and hurdle step patterns no sex differences were shown (*p* = 0.25–0.66). In addition, non-athletic boys showed a lesser number of asymmetries compared to non-athletic girls in the shoulder mobility test (50% vs. 67%, *df* = 1, *p* = 0.01), while other four uni/contralateral tests failed to reach significance (*p* = 0.06–0.4).

Among the subgroup of adolescents who have participated in sports, girls showed a lesser proportion of DFM compared to boys in inline lunge (13% vs. 31%, *df* = 1; *p* = 0.0009), shoulder mobility (20% vs. 51%, *df* = 1, *p* < 0.0001), and ASLR (11% vs. 28%, *df* = 1, *p* = 0.001). On the other hand, boys exhibited a lesser proportion of the DFM in the push-up test (43% vs. 85%, *df* = 1, *p* < 0.0001). However, there was no significant sex difference observed in squat, hurdle step, and rotary stability (*p* = 0.41–0.61). Concerning movement asymmetries, athletic girls demonstrated a lesser proportion of the asymmetries only in the ASLR movement pattern (13% vs. 22%, *df* = 1, *p* = 0.018).

## 4. Discussion

This study aimed to determine functional movement status in a general adolescent population. The main finding of this study is that adolescent boys showed a higher proportion of DFM and movement asymmetries in the larger number of FMS^TM^ tests compared to adolescent girls. More specifically, boys demonstrated a higher proportion of DFM and movement asymmetries in the inline lunge, shoulder mobility, and ASLR tests, which could potentially predispose them to higher injury for lower and upper extremities [5]. On the other hand, girls demonstrated a higher prevalence of DFM in the push-up and rotary stability tests. A low score in the trunk stability pushup test and rotary stability could indicate inadequate reactive stabilization of the trunk muscles and a deficit in the upper body strength in the female adolescent population [18]. For this reason, adolescent girls in the current study could be more prone to suffer from a higher risk of lower back injury [30]. On the other hand, girls slightly outperformed boys in total FMS^TM^ score (12.7 vs. 12.3 points) which further emphasized the aforementioned sex difference in functional movement during the mid-adolescent period.

In the current study, when the sample was stratified by sport participation, subgroup analyses showed similar results when compared to the findings of the initial analysis (i.e., for the total sample). The main difference between findings from the total sample and subsample was in the total FMS^TM^ score. More specifically, athletic girls outperformed athletic boys significantly (13.2 vs. 12.3, respectively), whereas in the subgroup of non-athletic participants sex difference was not noted. However, when the proportion of DFM and asymmetries are considered, similar patterns of movement dysfunction can be seen in both the athletic and non-athletic subgroup of participants, as well as within the total sample of mid-adolescents. This could possibly mean that sport participation probably does not influence functional movement status in the adolescent period since similar patterns of movement dysfunction were observed within the aforementioned groups and subgroups of mid-adolescents. Indeed, according to the current literature, in both the general and athletic adolescent population, most evidence demonstrates that females have a higher total FMS^TM^ score compared to males [8,9,10,12,15,16], although two studies reported opposite results [6,14]. In the study done by Abraham et al. [14], a large age span (10–17 yo) among participants revealed that pre-pubertal and pubertal subjects were included in the sample and all inactive children were excluded, which could potentially lead to higher mean values. Some researchers found no sex difference in total FMS^TM^ score [11,13], which could be potentially contributed to different populations studied (8-11 yo) and much smaller sample sizes (*n* = 77 and *n* = 58, respectively). Concerning individual FMS^TM^ patterns, evidence almost consistently shows that, in both general and athletic adolescents, same sex differences are present. More specifically, female adolescents generally show a better quality of movement in flexibility/mobility tests [10,11,13,16] while boys are better at push up and rotary stability [6,11,13,14,15,16]. Reported results from previous studies are in line with the findings of our study. What our study adds to the existing body of knowledge is that the same sex differences in functional movement exist in the population of mid-adolescents. 

Still, it remains unanswered as to why these sex differences in the functional movement patterns are present in the adolescent period. Therefore, three possible explanations for observed phenomena should be considered. *(1) Physiological*—*potential effect of maturation on muscle performance*: girls scored higher in the inline lunge, shoulder mobility, and ASLR, which could be due to higher mobility/flexibility demands of these movements [17,18]. This could be further explained with previous findings that reported greater mobility among girls compared to boys during the adolescent period of growth [31]. Since higher values of upper body strength are reported in boys compared to girls during adolescence [32], this could explain the discrepancy that was found in the upper body test (i.e., trunk stability push-up). *(2) Anatomical*—*potential effect of sex on joint morphology*: Reported differences in the aforementioned FMS^TM^ patterns could be possibly due to different architecture of the pelvis, hip, and shoulder since adolescent girls demonstrate more general joint laxity, hip anteversion, and tibiofemoral angles compared to adolescent boys [33]. Furthermore, development of the adolescent female pelvis from fifteen years of age and onward differs considerably from males, which can contribute to observed discrepancies in reported DFM in the current study [34]. The difference in the proportion of DFM in lower body patterns reported in the current study could be due to different hip architecture since it has been shown that adolescent girls have a different orientation of the acetabulum compared to boys [35]. More specifically, girls from the age of 13 to 17 have increased acetabular anteversion compared to boys [35]. This could possibly explain why girls performed better on tasks that demand active hip flexion (i.e., inline lunge and ASLR), whereas the different orientation of the acetabulum in boys could limit hip flexion movements. What could be concerning is that a higher prevalence of DFM observed in lower body patterns among boys could predispose them to a higher risk for developing hip orthopedic abnormalities (i.e., femoroacetabular impingement) [36]. *(3) Sociocultural—potential effect of cultural engagement in specific sport activity*: adolescent boys tend to engage more in sports such as soccer and basketball which have a high prevalence of unilateral and asymmetrical movement patterns [37]. This could further facilitate movement asymmetries seen in shoulder mobility and ASLR tests. On the other hand, girls participate more in sport activities that have an aesthetic component (i.e., dance, ballet, etc.) where specific unilateral movement patterns are not emphasized or trained in isolation [37]. Given the fact that in the current study more boys were engaged in sport activity compared to girls (48% vs. 25%, respectively), the aforementioned explanations could possibly explaine behind mechanism for observed discrepancies between adolescent girls and boys concerning movement asymmetries.

This study has several strengths. First, this is the only study that provides information about dysfunctional movement as well as movement asymmetries assessed by FMS^TM^ in a large sample of urban adolescents. Second, this is the first study to investigate a highly age-homogenized adolescent population (16–17 yo). Third, current research is based on a reasonably large number of participants (*n* = 733). All this allowed for more precise information about sex differences in a functional movement to be investigated. However, there are also several limitations that need to be considered while interpreting this data. This study investigated a population in the urban area, thus excluding children from rural areas which may affect the generalizability of the results in the context of the whole adolescent population. The large number of raters used in this study can be a potential drawback, although good interrater agreement in FMS^TM^ scores has been repeatedly reported [27,28]. Despite all this, the results of the present study give comprehensive data about a functional movement among the adolescent population.

## 5. Conclusions

The results of this study confirmed some previous findings and offer a new perspective in the context of functional movement in an adolescent population. In the current study, the total functional movement screen score was higher in girls compared to boys. Sex differences were present in several individual functional movement patterns, where boys demonstrated a higher prevalence of DFM in patterns that challenged mobility and flexibility of the body, while girls underperformed in tests that had higher demands for upper body strength and abdominal stabilization. The results of the present study need to be considered while implementing data into practical usage and while using FMS^TM^ as a screening tool among an adolescent school-aged population. Future research should focus on investigating sex dimorphism in functional movement in other populations of children and adolescents.

## Figures and Tables

**Figure 1 children-07-00308-f001:**
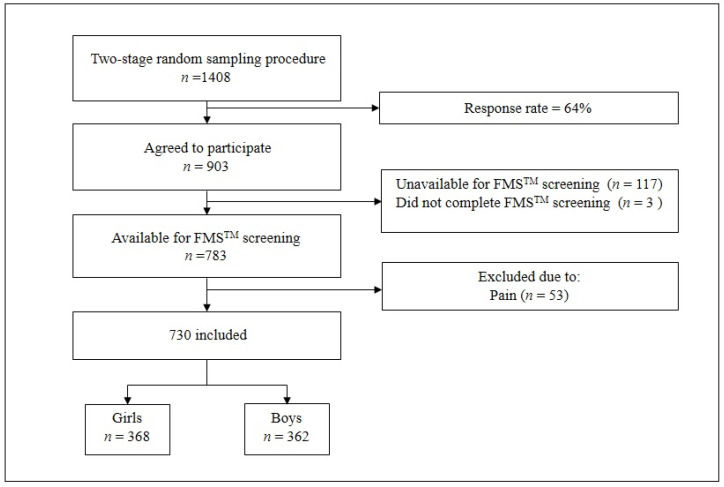
Flowchart of included participants.

**Figure 2 children-07-00308-f002:**
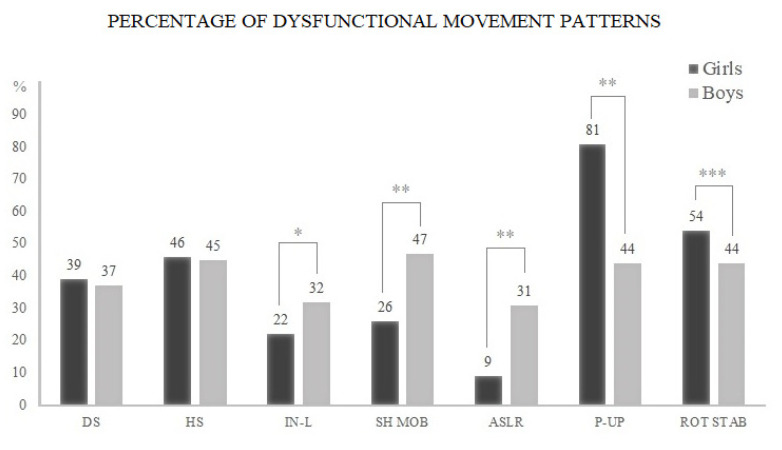
Proportion (%) of adolescent girls and boys who performed dysfunctional movement (DFM) in each Functional Movement Screen™ (FMS^TM^) test. Note: DS: deep squat; HS: hurdle step; IN-L: inline lunge; SHO MOB: shoulder mobility; ASLR: active straight leg raise; P-UP: Trunk stability push-up; ROT STAB: rotary stability. * *p* = 0.0009; ** *p* < 0.0001; *** *p* = 0.0075.

**Figure 3 children-07-00308-f003:**
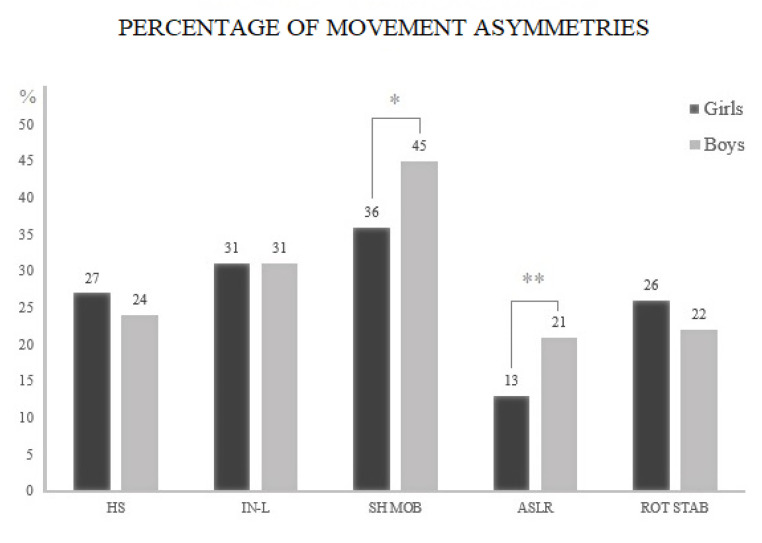
Proportion (%) of adolescent girls and boys who demonstrated movement asymmetries in each FMS^TM^ test. Note: HS: hurdle step; IN-L: inline lunge; SHO MOB: shoulder mobility; ASLR: active straight leg raise; ROT STAB: rotary stability. * *p* = 0.0218; ** *p* = 0.008.

**Table 1 children-07-00308-t001:** Basic characteristics of participants by sex.

Basic Characteristics		Girls	Boys
BMI (kg/m^2^) Mean (SD)		21.7 (3.2)	22.4 (3.5)
Waist Circumference (cm) Mean (SD)		68.7 (6.4)	76.0 (7.5)
Hips Circumference (cm) Mean (SD)		96.7 (7.5)	98.0 (7.5)
Sum of Four Skinfolds (mm) Mean (SD)		48.8 (15.0)	37.1 (18.1)
Functional Movement Asymmetries *n* (%)	0	76 (21)	86 (23)
	1	128 (35)	126 (34)
	2	98 (27)	111 (30)
	3	51 (14)	38 (10)
	4	7 (2)	7 (2)
	5	2 (0.5)	0 (0)
Sport Participation * *n* (%)		93 (25)	173 (48)
SES Median (IQR)		3 (1)	2 (1)

Note: BMI: Body Mass Index; Functional Movement Asymmetries *n* (%): Number (*n*) and percentage (%) of participants who exhibited Functional Movement Asymmetries within each sex group; Sport Participation *n* (%): Number (*n*) and percentage (%) of participants that participated in sport activity; SES: Socioeconomic status (1—Much lower than average, 2—Lower than average, 3—Average, 4—Higher than average, 5—Much higher than average); IQR: Interquartile Range; SD: Standard Deviation; * Data are presented for 725 participants.

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
