# Peer review of "Does Sex Dimorphism Exist in Dysfunctional Movement Patterns during the Sensitive Period of Adolescence?"

_children, 2020, doi:10.3390/children7120308_

Round 1

Reviewer 1 Report

Thank you for submitting your work to the journal. I have read the article with a great interest and admiration. 

This study aimed to investigate sex differences in the functional movement in the adolescent period. Seven hundred and thirty adolescents (365 boys) aged 16–17 years participated in the study. 

This paper is well-written and nicely organized. The findings are relevant and the methods were appropriately designed. The results are based on the findings. I believe the content and arguments of this study make due contributions to the ongoing development. 

Author Response

General comment to the Editor and Reviewers:

First of all, thank you for considering our manuscript for revisions to your journal of Children. All suggestions from the Editor and both Reviewers are accepted and incorporated into the manuscript. Also, content changes can be seen within the function ‘track changes’ while grammatical changes are not explicitly highlighted due to a number of errors through text which could impair the flow and readability of the text. In the revisited manuscript, the ‘Introduction’ section was improved (Please see pg.1.; lines: 35.-42.). Also, the ‘Methods’, ‘Results’ and ‘Discussion’ parts of the manuscript are changed and modified (Please see following: In ‘Materials’ and Methods’ section: pg.2.; lines: 84.-87.; pg.3. Figure 1.; pg.4. lines:130.-142.; In ‘Results’ section: lines: 179.-199.; In ‘Discussion’ section: lines: 214.-224.). Below are specific answers to the Editor and Reviewers comments.

Comments and Suggestions for Authors

Thank you for submitting your work to the journal. I have read the article with a great interest and admiration. 

This study aimed to investigate sex differences in the functional movement in the adolescent period. Seven hundred and thirty adolescents (365 boys) aged 16–17 years participated in the study. This paper is well-written and nicely organized. The findings are relevant and the methods were appropriately designed. The results are based on the findings. I believe the content and arguments of this study make due contributions to the ongoing development.

Response 1: Thank you for your nice words and provided interest in our paper. Nevertheless, we tried to improve our article with respect to your and other reviewers’ suggestions.

Additional response to the Reviewer’s suggestion:

Thank you for the suggestion regarding the introduction and its improvement. The authors of the current manuscript revisited the paper and improved the ‘Introduction’ section with more concise sentences (Please see pg.1.; lines: 35.-42.). Also, the ‘Methods’, ‘Results’ and ‘Discussion’ parts of the manuscript are improved (Please see following: In ‘Materials’  and Methods’ section: pg.2.; lines: 84.-87.; pg.3. Figure 1.; pg.4. lines:130.-142.; In ‘Results’ section: lines: 179.-199.; In ‘Discussion’ section: lines: 214.-224.). In addition, the authors wish to thank you for spotting language errors. Therefore, we made a grammatical correction along with proofreading through the whole manuscript. This suggestion improved our paper and the readability of the text.

Reviewer 2 Report

Why was one of the exclusion criteria that participants did not report any pain during movements? Precisely with the practice of exercise you can reduce and even eliminate pain from bad body postures or incorrect movement patterns. If possible, you explain this criterion better

The introduction talks about the health risk that physical inactivity entails, but neither the methodology nor the results take this parameter into account. Could you explain why?

Author Response

General comment to the Editor and Reviewers:

First of all, thank you for considering our manuscript for revisions to your journal of Children. All suggestions from the Editor and both Reviewers are accepted and incorporated into the manuscript. Also, content changes can be seen within the function ‘track changes’ while grammatical changes are not explicitly highlighted due to a number of errors through text which could impair the flow and readability of the text. In the revisited manuscript, the ‘Introduction’ section was improved (Please see pg.1.; lines: 35.-42.). Also, the ‘Methods’, ‘Results’ and ‘Discussion’ parts of the manuscript are changed and modified (Please see following: In ‘Materials’ and Methods’ section: pg.2.; lines: 84.-87.; pg.3. Figure 1.; pg.4. lines:130.-142.; In ‘Results’ section: lines: 179.-199.; In ‘Discussion’ section: lines: 214.-224.). Below are specific answers to the Editor and Reviewers comments.

Comments and Suggestions for Authors

Why was one of the exclusion criteria that participants did not report any pain during movements? Precisely with the practice of exercise you can reduce and even eliminate pain from bad body postures or incorrect movement patterns. If possible, you explain this criterion better

Response 1:  Thank you for the comment and for spotting this issue.  Also, the authors of this manuscript agree with the reviewers’ comment that exercise can reduce or eliminate pain. However, this paper deals with the dysfunctions in movement patterns not with the pain during movement. Since pain is multifactorial in its etiology (source can be from the physical, psychological, sociological component or combination; see Meints and Edwards, 2018), pain should be investigated separately from movement dysfunction. Since the aforementioned issue was not the aim of this study, we examined functional and dysfunctional movement patterns solely. In addition to the current study, several papers used similar or even identical criteria in the selection process (Karuc et al., 2020a, Karuc et al., 2020,b). Since pain present in the body can significantly influence the exhibition of optimal movement patterns (Sterling, et al., 2001), this exclusion criteria was mandatory. Indeed, evidence shows that pain can alter movement behavior (Sterling, et al., 2001).

The above-mentioned explanation is already mentioned in the originally submitted manuscript under the ’2.1. Participants’ section, where following was stated: ‘All the participants had to meet certain criteria for the medical doctor to perform the screening process, specifically: 1) not having any pain during the movement screening (i.e. FMSTM testing procedures)..’ (please see original manuscript, pg.2. lines: 81.-83.). Also, in the ‘2.2. Functional Movement Screen‘ authors explicitly explained the reason why this criterion was used: ‘Evidence shows that pain can alter movement control [30]. Therefore, subjects were asked if they felt pain during the FMSTM assessment, and were subsequently scored with the score of 0 and excluded if answered positively to this question (n=53).’ (please see original manuscript, pg.3.; lines: 105.-107.).

References for Response 1:

Meints, S. M., & Edwards, R. R. (2018). Evaluating psychosocial contributions to chronic pain outcomes. Progress in neuro-psychopharmacology & biological psychiatry, 87(Pt B), 168–182. https://doi.org/10.1016/j.pnpbp.2018.01.017

Sterling, M., Jull, G., & Wright, A. (2001). The effect of musculoskeletal pain on motor activity and control. The journal of pain : official journal of the American Pain Society2(3), 135–145. https://doi.org/10.1054/jpai.2001.19951

Karuc, J., Mišigoj-Duraković, M., Marković, G., Hadžić, V., Duncan, M. J., Podnar, H., & Sorić, M. (2020). Movement quality in adolescence depends on the level and type of physical activity. Physical therapy in sport 46, 194–203. https://doi.org/10.1016/j.ptsp.2020.09.006

Karuc, J.; Marković, G.; Mišigoj-Duraković, M.; Duncan, M.J.; Sorić, M. Is Adiposity Associated with the Quality of Movement Patterns in the Mid-Adolescent Period? Int. J. Environ. Res. Public Health 2020, 17, 9230. https://doi.org/10.3390/ijerph17249230

The introduction talks about the health risk that physical inactivity entails, but neither the methodology nor the results take this parameter into account. Could you explain why?

Response 2: Thank you for this important note. Authors of the current manuscript agree with the reviewers’ comment that the introduction part talks about health risk while neither the methodology nor the results take this into account. Since physical inactivity is related to a higher risk of morbidity and other cardiometabolic risks, and positively related to functional movement, the authors of the current manuscript considered that this introduction was appropriate. Since the aim of the current paper was to examine sex dimorphism in functional movement patterns and movement asymmetries in the adolescent population (please see pg.2; lines: 71-72), there was no need to mention this in the ‘Methods’ or ‘Results’ section. However, the authors of the current manuscript reframed the initial part of the ‘Introduction’ section as suggested by Reviewer 1 and Reviewer 2 in order to get a more concise text (Please see pg.1.; lines: 33.-42).

Reviewer 3 Report

Title: Does sex dimorphism exist in dysfunctional movement patterns during sensitive period of adolescence?

In this study, the authors investigated sex differences in the functional movement in the adolescent period. This study has chosen a standardized Functional Movement Screen™ (FMSTM) protocol for a sex difference in the functional movement. However, there are some comments to be addressed.

  • In general, FMS uses cut-off scores. Why did you use DFM?
  • A score of <=14 on the FMS is used as the cut-off score.
  • FMS predicts the possibility of functional limitations or injuries with cut-off scores. In what perspective was the cut-off score used in this study?
  • Only scores corresponding to the cut-off score were presented in the abstract and discussion.
  • Even if DFM is used, it seems that the results should be analyzed with FMS scores.
  • A lower FMS scores have been noted to be associated with an increased BMI, increased age, and decreased activity level (Mitchell UH, Johnson AW, Vehrs PR, Feland JB, & Hlton SC. Performance on the Functional Movement Screen in older active adults. J Sport & Health Science. 2016; 5(1): 119-125.)
  • Line 79-80: One hundred and twenty participants were unavailable on the day of testing or did not complete the FMS screening.
  • Why did the subject do not complete the FMS screening?
  • This reason seems to be important for the functional evaluation of adolescents.
  • The sport participation was 93 for girls and 173 for boys, and it seems that whether or not they participated in sports acted as a bias to DFM. I think it is necessary to separately analyze only the subjects who participated in the sport.
  • Is there a reason for comparing only 5 of the 7 items of FMS in Figure 3?
  • Line 161-165. In this study, there were differences in 2 for men and 3 for women (Table 2). These results have already suggested differences in the three items (inline lunge, shoulder mobility, and ASLR tests) in men in the previous study. Is there any reason for re-study?
  • Line 169-172. On the other hand, girls slightly outperformed boys in total FMS score (12.7 vs. 12.3 points) which further emphasizes the aforementioned sex difference in the functional movement during the mid-adolescent period.
  • In what perspective was the cut-off score used in this study?
  • Modify the discussion.

Author Response

General comment to the Editor and Reviewers:

First of all, thank you for considering our manuscript for revisions to your journal of Children. All suggestions from the Editor and both Reviewers are accepted and incorporated into the manuscript. Also, content changes can be seen within the function ‘track changes’ while grammatical changes are not explicitly highlighted due to a number of errors through text which could impair the flow and readability of the text. In the revisited manuscript, the ‘Introduction’ section was improved (Please see pg.1.; lines: 35.-42.). Also, the ‘Methods’, ‘Results’ and ‘Discussion’ parts of the manuscript are changed and modified (Please see following: In ‘Materials’ and Methods’ section: pg.2.; lines: 84.-87.; pg.3. Figure 1.; pg.4. lines:130.-142.; In ‘Results’ section: lines: 179.-199.; In ‘Discussion’ section: lines: 214.-224.). Below are specific answers to the Editor and Reviewers comments.

Comments and Suggestions for Authors

Title: Does sex dimorphism exist in dysfunctional movement patterns during sensitive period of adolescence?

In this study, the authors investigated sex differences in the functional movement in the adolescent period. This study has chosen a standardized Functional Movement Screen™ (FMSTM) protocol for a sex difference in the functional movement. However, there are some comments to be addressed.

Response to general comment: Thank you for your overall comment about our study.

Comment 1: In general, FMS uses cut-off scores. Why did you use DFM? A score of <=14 on the FMS is used as the cut-off score.

Response 1:

Thank you very much for this comment. We agree with the reviewers’ comment that the FMSTM diagnostic instrument uses cut-off scores (<=14 on the FMS continuous scale) as the number of researches utilized this criterion. However, in the previous studies (Dossa et al., 2014; Bardenett et al., 2015; Warren et al., 2015), a cut-off score <=14 was used for the sum of all 7 individual FMSTM tests whereas, in the current study, DFM was used for the individual FMSTM patterns.  What is more, a large body of evidence does not support the FMSTM cut-off score of < =14 as sufficient to predict the injury occurrence (please see Response 2 for detailed explanations on this topic). The authors of this research used the dysfunctional movement (DFM) category as the qualitative representation of the non-optimal movement quality for individual FMS movement patterns. Therefore, in order to better describe what dysfunctional movement is, the above-mentioned criterion was used. This was further described in the originally submitted manuscript (please see in the original manuscript, pg.3.; lines:107.-114.). To support the aforementioned explanation, several papers used the same criteria as proposed in the current study. Nicolozakes et al., (2018) describe this criterion as well:

''We operationally defined a cut-off score of ≤1 as poor performance on an individual FMS™ test because a score of ≤1 indicates that a subject is unable to perform the movement or assume the position of an individual FMS™ test. The cut-off score of ≤1 has previously been used when analyzing individual test scores.''

Also, several other authors utilized the same categorization for the individual FMSTM movement patterns. Duncan et al. pointed out in their findings:

''However, the pattern of scoring differed depending on the particular test within the FMS between weight status groups. In all cases normal-weight children were more likely to score a ‘2’ or a ‘3’ than their obese peers (all P = .005 or better). However, normal weight children only scored significantly better than overweight children for the deep squat (P = .0001) and shoulder mobility tests (P = .04).’’

In addition, the authors of the current study proposed this criterion in the recently published article titled: ‘Is Adiposity Associated with the Quality of Movement Patterns in the Mid-Adolescent Period?’ (Karuc et al., 2020):

'In this study, we defined FM as the movement with a given score of 2 or 3 during the FMSTM procedure. Furthermore, a score of 1 was given when the participant was unable to perform movement due to the number of movement compensation present which reflects the DFM pattern [26,27,28]. This means that score of 2 and 3 was an indicator of FM, whereas a score of 1 was an indicator of DFM for each of 7 individual FMS tests. In this way, we could calculate the number and proportion of participants that exhibited DFM in each of the 7 individual FMS tests. This was the basic step for analyzing the differences in the proportion of participants that performed DFM between normal weight and overweight children for each of 7 individual FMSTM tests (i.e., using chi-square tests). Besides, an overall composite score (total FMS score) was calculated with a total FMS score of 21 according to standardized guidelines reported in the literature [26,27].'

References for Response 1:

Dossa, K.; Cashman, G; Howitt, S.; West, B.; Murray, N. Can injury in major junior hockey players be predicted by a pre-season functional movement screen – a prospective cohort study. J. Can. Chiropr. Assoc. 2014, 58, 421-427.

Bardenett, S.M.; Micca, J.J.; DeNoyelles, J.T.; Miller, S.D.; Jenk, D.T.; Brooks, G.S. Functional Movement Screen Normative Values and Validity in High School Athletes: Can the FmsTM Be Used As a Predictor of Injury? Int. J. Sports Phys. Ther. 2015, 10, 303–308.

Warren, M., Smith, C. A., & Chimera, N. J. (2015). Association of the Functional Movement Screen with injuries in division I athletes. Journal of sport rehabilitation24(2), 163–170. https://doi.org/10.1123/jsr.2013-0141

Dorrel, B. S., Long, T., Shaffer, S., & Myer, G. D. (2015). Evaluation of the Functional Movement Screen as an Injury Prediction Tool Among Active Adult Populations: A Systematic Review and Meta-analysis. Sports health7(6), 532–537. https://doi.org/10.1177/1941738115607445

Moran, R. W., Schneiders, A. G., Mason, J., & Sullivan, S. J. (2017). Do Functional Movement Screen (FMS) composite scores predict subsequent injury? A systematic review with meta-analysis. British journal of sports medicine51(23), 1661–1669. https://doi.org/10.1136/bjsports-2016-096938

Nicolozakes, C. P., Schneider, D. K., Roewer, B. D., Borchers, J. R., & Hewett, T. E. (2018). Influence of Body Composition on Functional Movement Screen™ Scores in College Football Players. Journal of sport rehabilitation27(5), 431–437. https://doi.org/10.1123/jsr.2015-0080

Duncan, M. J., Stanley, M., & Leddington Wright, S. (2013). The association between functional movement and overweight and obesity in British primary school children. BMC sports science, medicine & rehabilitation5, 11. https://doi.org/10.1186/2052-1847-5-11

Karuc, J.; Marković, G.; Mišigoj-Duraković, M.; Duncan, M.J.; Sorić, M. Is Adiposity Associated with the Quality of Movement Patterns in the Mid-Adolescent Period? Int. J. Environ. Res. Public Health 2020, 17, 9230.)

 Comment 2: FMS predicts the possibility of functional limitations or injuries with cut-off scores. In what perspective was the cut-off score used in this study? Only scores corresponding to the cut-off score were presented in the abstract and discussion. Even if DFM is used, it seems that the results should be analyzed with FMS scores. A lower FMS scores have been noted to be associated with an increased BMI, increased age, and decreased activity level (Mitchell UH, Johnson AW, Vehrs PR, Feland JB, & Hlton SC. Performance on the Functional Movement Screen in older active adults. J Sport & Health Science. 2016; 5(1): 119-125.

Response 2: Thank you for pointing out this issue. Concerning the usage of the FMSTM as the injury prediction tool, a large body of evidence do not support FMSTM as an injury prediction instrument (Dossa et al., 2014; Bardenett et al., 2015; Warren et al., 2015). Also, the proposed FMSTM cut-off score of <=14 is not supported within the scientific body of knowledge. Indeed, several recently published systematic reviews and meta-analyses do not support usage of the FMSTM with the associated criterion of <=14 for injury prediction (Dorrel et al., 2015, Moran et al., 2017). In addition, authors of the current article reported poor discriminatory ability of the FMSTM to detect injury occurrence in the representative sample of urban adolescents even with the usage of more sophisticated methodology (artificial intelligence and machine learning) (the article is currently in the second round of the revision process; submitted in the Journal of Strength and Conditioning Research). Therefore, for the above-mentioned reasons, the FMSTM cut-off score of 14 points was not used in this study. Another reason is that the aim of the current manuscript was not to investigate FMSTM as an injury prediction tool indeed (please see in the original manuscript, pg.2.; lines:67.-68.).

Also, the authors wish to thank you for the question regards to perspective of the cut-off score used in this study: ‘In what perspective was the cut-off score used in this study?‘. As stated in the previous Response (i.e. Response 1), the only cut-off score that was used in the current manuscript was a cut-off score of =1 as the indicator of DFM (i.e. poor performance on an individual FMS™ test) where a score above 1 (i.e. score of 2 and 3) was an indicator of better/optimal movement quality (i.e. functional movement - FM). This criterion is supported through a number of studies. 

Authors of the current manuscript appreciate the second part of the reviewers’ comment: ‘Even if DFM is used, it seems that the results should be analyzed with FMS scores.’. However, in the current study, usage of the above-mentioned criterion for dysfunctional movement (DFM) was supported with the previous evidence as noted in the previous response (please see Response 1). In addition, several papers used this criterion for differentiation between functional and dysfunctional (DFM) movement patterns assessed via FMSTM (Nicolozakes et al, 2018; Duncan et al., 2013; Karuc et al, 2020).

The authors of this manuscript agree with the reviewers’ comment that lower FMS scores have been noted to be associated with an increased BMI, increased age, and decreased activity level. Authors of the current manuscript are aware of the potential influence of other factors that can influence FMSTM results since the same group of authors have already published an article that deals with BMI (and adiposity), physical activity level, maturation, etc. (Karuc and Mišigoj-Duraković, 2109, Karuc et al., 2020a, Karuc et al., 2020b). However, this comment does not lie within the scope of the current article since the aim of this paper was to examine sex dimorphism in the population of urban adolescents.

References:

Dossa, K.; Cashman, G; Howitt, S.; West, B.; Murray, N. Can injury in major junior hockey players be predicted by a pre-season functional movement screen – a prospective cohort study. J. Can. Chiropr. Assoc. 2014, 58, 421-427.

Bardenett, S.M.; Micca, J.J.; DeNoyelles, J.T.; Miller, S.D.; Jenk, D.T.; Brooks, G.S. Functional Movement Screen Normative Values and Validity in High School Athletes: Can the FmsTM Be Used As a Predictor of Injury? Int. J. Sports Phys. Ther. 2015, 10, 303–308.

Warren, M., Smith, C. A., & Chimera, N. J. (2015). Association of the Functional Movement Screen with injuries in division I athletes. Journal of sport rehabilitation24(2), 163–170. https://doi.org/10.1123/jsr.2013-0141

Dorrel, B. S., Long, T., Shaffer, S., & Myer, G. D. (2015). Evaluation of the Functional Movement Screen as an Injury Prediction Tool Among Active Adult Populations: A Systematic Review and Meta-analysis. Sports health7(6), 532–537. https://doi.org/10.1177/1941738115607445

Moran, R. W., Schneiders, A. G., Mason, J., & Sullivan, S. J. (2017). Do Functional Movement Screen (FMS) composite scores predict subsequent injury? A systematic review with meta-analysis. British journal of sports medicine51(23), 1661–1669. https://doi.org/10.1136/bjsports-2016-096938

Nicolozakes, C. P., Schneider, D. K., Roewer, B. D., Borchers, J. R., & Hewett, T. E. (2018). Influence of Body Composition on Functional Movement Screen™ Scores in College Football Players. Journal of sport rehabilitation27(5), 431–437. https://doi.org/10.1123/jsr.2015-0080

Duncan, M. J., Stanley, M., & Leddington Wright, S. (2013). The association between functional movement and overweight and obesity in British primary school children. BMC sports science, medicine & rehabilitation5, 11. https://doi.org/10.1186/2052-1847-5-11

Karuc, J.; Mišigoj-Duraković, M. Relation between Weight Status, Physical activity, Maturation, and Functional Movement in Adolescence: An Overview. J. Funct. Morphol. Kinesiol. 2019, 4, 31. https://doi.org/10.3390/jfmk4020031

Karuc, J.; Marković, G.; Mišigoj-Duraković, M.; Duncan, M.J.; Sorić, M. Is Adiposity Associated with the Quality of Movement Patterns in the Mid-Adolescent Period? Int. J. Environ. Res. Public Health 2020, 17, 9230.)

Karuc, J., Mišigoj-Duraković, M., Marković, G., Hadžić, V., Duncan, M. J., Podnar, H., & Sorić, M. (2020). Movement quality in adolescence depends on the level and type of physical activity. Physical therapy in sport46, 194–203. https://doi.org/10.1016/j.ptsp.2020.09.006

Comment 3: Line 79-80: One hundred and twenty participants were unavailable on the day of testing or did not complete the FMS screening. Why did the subject do not complete the FMS screening? This reason seems to be important for the functional evaluation of adolescents.

Response 3: Thank you for this comment and for stressing this issue.  We agree with the reviewer’s suggestion that the reason for unavailability for FMSTM screening is important to report within the study.

Of one hundred and twenty participants, one hundred and seventeen were unavailable on the day of testing because they were missing from the school at the time of the measurements whereas three subjects did not complete FMSTM screening due to lack of time (1 girl and 2 boys). In line with this, the authors incorporated an explanation in the revisited manuscript regarding the above-mentioned issue (please see in the revisited manuscript, pg.2.; lines: 84.-87.). Additionally, information within the flowchart was corrected and previous Figure 1 replaced with the new one (please see Figure 1., pg.3.).

Comment 4: The sport participation was 93 for girls and 173 for boys, and it seems that whether or not they participated in sports acted as a bias to DFM. I think it is necessary to separately analyze only the subjects who participated in the sport.

Response 4: Thank you for this very important comment and suggestion. The authors of the current manuscript agree with the reviewers’ opinion that whether or not subjects participated in sports could act as a bias to DFM. The authors accepted the suggestion that analysis should be done separately for each subgroup (athletic/non-athletic participants). Therefore, the authors made additional analysis and incorporated the findings into the ‘Results’ section of the revisited paper (additional paragraph was added in the revisited manuscript, please see pg. 6; lines: 179.-199.).

Comment 5: Is there a reason for comparing only 5 of the 7 items of FMS in Figure 3?

Response 5: Thank you for this question. Since 5 of 7 FMSTM items are contr/unilateral in their nature (ie. (i.e. hurdle step, inline lunge, shoulder mobility, active straight leg raise (ASLR), and rotary stability)., only for those movement patterns asymmetry can be observed. Since Figure demonstrates sex-differences in the movement asymmetries, for the above-mentioned reason, five of seven items were shown in Figure 3. Indeed, according to the literature,  only 5 of 7 FMSTM  items assess movement asymmetries (the difference between scores on the left and right side) as documented in the papers done by Cook et al. (2006a, 2006b). In line with this,  in the originally submitted manuscript following was stated: ‘’…FMSTM can detect movement asymmetries if the difference between right and left side of the uni/contralateral movement patterns is observed.. ‘’ (please see ‘Introduction section’ of the original manuscript, pg.2.; lines: 50.-52.). Also, the authors pointed out: ‘’We analyzed movement asymmetries for five contra/unilateral FMSTM tests (i.e. hurdle step, inline lunge, shoulder mobility, ASLR, and rotary stability).’’ (please see the section of the original manuscript: ’’2.2. Functional Movement Screen ‘’, pg.3.; lines: 114.-116.). For a detailed description of the FMSTM procedures please see a section of the original manuscript: ‘2.2. Functional Movement Screen’ (pg.3.; lines: 96.-121.).

References for Response 5:

Cook, G.; Burton, L.; Hoogenboom, B. Pre-Participation Screening: The Use of Fundamental Movements As An Assessment of Function–Part 1. North Am. J. Sport. Phys. Ther. 2006, 1, 62-72. doi: 10.1055/s-0034-1382055.

Cook, G.; Burton, L.; Hoogenboom, B. Pre-Participation Screening: The Use of Fundamental Movements As An Assessment of Function–Part 2. North Am. J. Sport. Phys. Ther. 2006, 1, 62, 132-139. doi: 10.1055/s-0034-1382055.

Comment 6: Line 161-165. In this study, there were differences in 2 for men and 3 for women (Table 2). These results have already suggested differences in the three items (inline lunge, shoulder mobility, and ASLR tests) in men in the previous study. Is there any reason for re-study?

Response 6: Thank you very much for this important note. The authors of the current study agree with the reviewers’ comment that similar differences in FMSTM tests were reported in the previous studies. Since previous studies were either small scale studies, included only active adolescents, or investigated a special population of adolescents (i.e. adolescents with overweight/obesity, etc.), this study was needed (as stressed in the Introduction; please see pg. 1.; lines: 64.-68).  What this study adds on the previous scientific knowledge is that aforementioned differences are present in the population of the average adolescents aged between 16-17 yo. Also, information about discrepancies in movement asymmetries further emphasizes sex dimorphism. Additionally, the main reason for re-study is that this is the first time that these difference were presented in a highly age-homogenized adolescent population (16-17 y) with a reasonably large number of participants (n=733), which was emphasized already in the originally submitted manuscript (please see ‘Conclusion’ part of the original manuscript, lines: 220.-225.). This allows more precise information about sex differences in functional movement which had not been investigated before with the aforementioned methodology.

Comment 7: Line 169-172. On the other hand, girls slightly outperformed boys in total FMS score (12.7 vs. 12.3 points) which further emphasizes the aforementioned sex difference in the functional movement during the mid-adolescent period. In what perspective was the cut-off score used in this study? Modify the discussion.

Response 7: Thank you for the overall comment about the discussion part of the manuscript. The authors appreciate this suggestion. The response for this comment (i.e. about the cut-off score) is already given within the previous answers (please see Response 1 and Response 2).

The authors accept the reviewers’ suggestions to modify the discussion. Therefore, the discussion part was modified according to this and previous reviewers’ comments (please see lines: 213.-224.).

Round 2

Reviewer 3 Report

Dear Sir

The author has improved the manuscript according to received comments.

And it is very well explained and interesting.
Thanks.
Lee